# Ecophysiology of Antarctic Vascular Plants: An Update on the Extreme Environment Resistance Mechanisms and Their Importance in Facing Climate Change

**DOI:** 10.3390/plants13030449

**Published:** 2024-02-03

**Authors:** Constanza F. Ramírez, Lohengrin A. Cavieres, Carolina Sanhueza, Valentina Vallejos, Olman Gómez-Espinoza, León A. Bravo, Patricia L. Sáez

**Affiliations:** 1Laboratorio Cultivo de Tejidos Vegetales, Centro de Biotecnología, Facultad de Ciencias Forestales, Universidad de Concepción, Concepción 4030000, Chile; consramirez@udec.cl (C.F.R.); valevallejosaravena@gmail.com (V.V.); 2Instituto de Ecología y Biodiversidad-IEB, Las Palmeras 3425, Ñuñoa, Santiago 7800003, Chile; lcaviere@udec.cl; 3ECOBIOSIS, Departamento de Botánica, Facultad de Ciencias Naturales y Oceanográficas, Universidad de Concepción, Barrio Universitario s/n, Concepción 4030000, Chile; 4Laboratorio de Fisiología Vegetal, Departamento de Botánica, Facultad de Ciencias Naturales y Oceanográficas, Universidad de Concepción, Barrio Universitario s/n, Concepción 4030000, Chile; csanhuez@udec.cl; 5Laboratorio de Fisiología y Biología Molecular Vegetal, Departamento de Ciencias Agronómicas y Recursos Naturales, Facultad de Ciencias Agropecuarias y Medioambiente, Universidad de La Frontera, Temuco 4811230, Chile; olman.gomez@ufrontera.cl (O.G.-E.) leon.bravo@ufrontera.cl (L.A.B.)

**Keywords:** Antarctic plant species, ecophysiological traits, climate change, *Colobanthus quitensis*, *Deschampsia antarctica*

## Abstract

Antarctic flowering plants have become enigmatic because of their unique capability to colonize Antarctica. It has been shown that there is not a single trait that makes *Colobanthus quitensis* and *Deschampsia antarctica* so special, but rather a set of morphophysiological traits that coordinately confer resistance to one of the harshest environments on the Earth. However, both their capacity to inhabit Antarctica and their uniqueness remain not fully explained from a biological point of view. These aspects have become more relevant due to the climatic changes already impacting Antarctica. This review aims to compile and update the recent advances in the ecophysiology of Antarctic vascular plants, deepen understanding of the mechanisms behind their notable resistance to abiotic stresses, and contribute to understanding their potential responses to environmental changes. The uniqueness of Antarctic plants has prompted research that emphasizes the role of leaf anatomical traits and cell wall properties in controlling water loss and CO_2_ exchange, the role of Rubisco kinetics traits in facilitating efficient carbon assimilation, and the relevance of metabolomic pathways in elucidating key processes such as gas exchange, nutrient uptake, and photoprotection. Climate change is anticipated to have significant and contrasting effects on the morphophysiological processes of Antarctic species. However, more studies in different locations outside Antarctica and using the latitudinal gradient as a natural laboratory to predict the effects of climate change are needed. Finally, we raise several questions that should be addressed, both to unravel the uniqueness of Antarctic vascular species and to understand their potential responses to climate change.

## 1. Introduction

Since Skottsberg [1], scientists have been intrigued about why *Deschampsia antarctica* and *Colobanthus quitensis* are the only vascular plant species that naturally colonized and thrive on the environmental conditions that characterize Antarctica, what is so special in these two species, how unique they are, and why only these two have colonized Antarctica. Although the mechanisms deployed by the Antarctic plants (see [2,3,4,5] and references therein) are also present in other plant species inhabiting hostile cold climates, what really distinguishes them from other cold-resistant plants is that they must cope with constant low temperatures during the whole day and the entire snow-free period [6]. This imposes a major constraint: they must grow and reproduce under continuous low-temperature conditions. Thus, a complex arrangement of traits, such as xerophytism, tolerance to low temperature, ability to maintain positive net photosynthesis at low temperature, adequate management of excess irradiance, and tolerance to water stress, are coordinated with compensatory mechanisms to deal with the constant low temperatures. For instance, high values of stomatal conductance compensate for the very low mesophyll conductances for CO_2_ (g_m_) found in these species [7]. The low g_m_, which limits the CO_2_ for carbon assimilation (A_N_), is compensated by a Rubisco enzyme highly specific for CO_2_ [7]. Further, all these traits are strongly coordinated with leaf hydraulic properties [8] to be highly efficient in water transport but, at the same time, minimize the risks of water transport collapse due to freezing temperatures. Although all these traits have shed light on the uniqueness of the Antarctic species, their capacity to inhabit Antarctica remains not fully explained from a biological point of view. For instance, factors that promote/restrict their distribution at different spatial scales, the genetic basis associated with stress resistance, biochemical traits such as saturated and unsaturated fatty acid ratio, and the production and allocation of nonstructural carbohydrates are among the biological aspects not yet fully studied in these plant species. In addition, along the high-elevation habitats of the Andes, it is possible to find habitats as extreme as Antarctica, showing a range of stressful conditions (i.e., extreme low temperatures, limited liquid water availability, high radiation, poorly developed soils, etc.). In these habitats, it is possible to find populations of the Antarctic vascular species [9]: *C. quitensis* grows from Mexico to the south of the Antarctic Peninsula (along the Andes), and *D. antarctica* is distributed along the Andes from 33° S latitude to Antarctica [9]. However, there are few comparative studies between Andean high-elevation habitats and Antarctic populations that can shed light on the uniqueness of the ecophysiological traits that are displayed under Antarctic conditions. Studies of latitudinal gradients are essential to elucidate plant responses to a broader range of both abiotic and biotic environmental conditions and could be used as an indirect predictor of climate change.

Maritime Antarctica and the Antarctic Peninsula are among the world regions that have experienced the strongest warming trends [10,11]. The manifestation of climate warming in these areas is evident in the significant retreat of glaciers and ice shelves. These physical changes in space, together with longer growing seasons, higher temperatures, and rainfall frequencies, have provided new areas available for colonization and spread of plant populations [12]. Consequently, increases in population size and reproduction of the Antarctic vascular plants leading to an increase in the expansion range of both species have been reported [13,14]. Apparently, a small increase in air temperature may be beneficial for a variety of physiological processes (e.g., photosynthesis). However, more frequent leaf temperatures over 20 °C could be harmful, affecting photosynthesis and reducing the ability of these plants to tolerate freezing temperatures and activate photoprotection mechanisms at low temperatures, which are attributes by which they are able to colonize these very harsh environments [5,15]. Moreover, several studies have reported that both species are being affected differently by warming in Maritime Antarctica [16,17,18].

In our previous review [5], several questions were raised about the ecophysiology of Antarctic vascular plants, their adaptative mechanisms, and their responses to future climate change. Some of these questions have been answered in recent years, but others remain elusive. The objective of this review is to report the recent advances in the ecophysiology of Antarctic vascular plants, in particular: (1) to deepen and update the mechanisms that underlie the ability of these plant species to resist environmental stresses, (2) to gather information regarding their performance throughout their natural distribution in Antarctica and out of it, and (3) to inquire about the responses of their adaptative mechanisms in facing climate change.

## 2. Ecophysiological Adaptations of Antarctic Vascular Plants

### 2.1. Freezing Resistance

Several studies have evaluated the freezing resistance of both Antarctic species (see [5] and Table 1). However, due to only laboratory-grown plants having been evaluated, the freeze tolerance of Antarctic species could be underestimated. Sierra-Almeida and colleagues [18] carried out an *in situ* study of the lethal temperature for 50% of leaf tissues (LT_50_) at three contrasting sites on King George Island, differing in soil nutrients, plant cover, and relative abundance of the Antarctic vascular plant species. The average LT_50_ observed was −22.8 °C for *D. antarctica* and −15.3 °C for *C. quitensis*, agreeing with those observed in whole plants without nucleating agents by Gianoli et al. [19]. Additionally, at all sites, the nucleation temperature was higher than LT_50_ in both Antarctic species, suggesting that both species can tolerate ice formation within their leaf tissues. Therefore, under Antarctic field conditions, both *C. quitensis* and *D. antarctica* are freezing-tolerant plant species [18].

The accumulation and/or activity of certain cryoprotective proteins in cold-exposed plants is crucial to increasing membrane stability and preventing damage caused by extracellular ice growth and propagation. *D. antarctica* presents a high constitutive activity of antifreeze proteins in the apoplast [24], as well as stress-related proteins such as dehydrins [25]. Both groups of proteins may play an important role in cryoprotection and in the prevention of freeze-induced cell dehydration, which is consistent with the greater freezing tolerance exhibited by *D. antarctica*. Surprisingly, *C. quitensis* has shown no antifreeze activity [24] and has very low recrystallization inhibition activity [26]. In addition, the production of low molecular weight cryoprotectants has also been reported to play a key role in regulating the osmotic balance between cell compartments and membrane stabilization [27].

Different studies have reported high carbohydrate concentrations in the leaves of both Antarctic vascular species, mainly soluble sugars [28,29]. In particular, at low temperatures (4 °C), both Antarctic species store high amounts of sucrose [21,30]. Additionally, both species contain polymers of (fructosyl)n-sucrose. During cold acclimation, *C. quitensis* accumulates oligosaccharides of the raffinose family (RFO), whilst *D. antarctica* stores fructans [21,28,29]. Thus, the sugars stored in the Antarctic plants could be useful during periods of negative carbon balance (i.e., when the respiratory rate exceeds the photosynthetic rate due to a metabolic shift induced by environmental stressors). Sucrose accumulation and allocation are regulated by day length and growth temperature [30,31]. In long daylight (photoperiod: 21/3 h day/night; LDL) and low temperature (4 °C), sucrose accumulation is related to an increase in the sugar export from the crown in *D. antarctica* and from the root system in *C. quitensis* [3,30,31]. In plants of *D. antarctica* growing at 16/8 h photoperiod (medium daylight; MDL), whilst sucrose content increased in leaves, it decreased in non-photosynthetic organs. Thus, it seems that in early summer with long daylight (LDL), *D. antarctica* increases sucrose synthesis in the crown and is allocated to other organs, while later, during the growing season, there is the conversion of sucrose to other sugars or its degradation [30]. This agrees with Bravo et al. [21], who reported that under MDL, high fructans accumulation occurs. In the field, when the daylight decreased to 8 h (close to March), *D. antarctica* probably stored part of the carbohydrates in the crown. Regarding *C. quitensis*, Zuñiga-Feest et al. [31] reported an increment in sucrose content when grown at 4 °C and in short daylight (photoperiod: 8/16 h day/night). These authors suggest that the accumulation of sugar could be caused by the reduction in growth with the concomitant reduction in the use of carbon skeletons as an energy source or with an increase in the biosynthesis of sucrose when detecting the shortening of day length that occurs at the end of the growing season in Maritime Antarctica.

On the other hand, the high sucrose concentration in both Antarctic species has been related to the high activity of the enzyme sucrose phosphate synthase (SPS, [30]), which is regulated by light and cold temperatures [30,32,33]. Cold acclimation induces an increase in SPS activity without an increase in protein levels [32,33]. This phenomenon has been reported for *C. quitensis* (for more details, see [3]) and suggests that a high SPS activity during the Antarctic summer can provide the necessary carbohydrates to supply rapid vegetative and reproductive growth [31]. Despite the great progress on this subject, the carbohydrate store, their allocation during the growing season, their use during vernalization, and the translocation in response to freeze-thaw events in the field are issues that have not yet been studied.

Acclimation to freezing temperatures is a genetically complex process involving a wide range of physiological adjustments that are largely based on complex changes in signal transductions and gene expressions [34]. It has been shown that the “C repeat binding factor” (CBF or CBF/DREB1) transcription factors are key in the regulation of genes implied in plant freezing tolerance. These genes additionally include other genes like transcription factors, channel proteins, and membrane transporters, enzymes related to sugar and proline metabolism, etc. [35,36,37]. Recently, in a laboratory study, López et al. [38] evaluated for the first time the gene expression of cold-acclimated *D. antarctica* plants. They reported that cold-acclimated plants showed significant enrichment in genes related to transporter activity and membrane structure, such as UDP-glycosyltransferase activity. Among the genes related to cold tolerance, the following are noteworthy: CBF or CBF/DREB1, sugar metabolism, dehydrins, and other proteins related to protection. Clemente-Moreno et al. [39,40] reported significant increases in metabolites related to osmoprotection and membrane stabilization at low temperatures (proline, sucrose, fructose, galactinol, raffinose, and GABA) in both Antarctic species. All these reports constitute a new insight that helps to explain the remarkable freezing resistance of Antarctic vascular plants.

### 2.2. Photosynthesis

There is consensus that both Antarctic species rely on a robust CO_2_ assimilation machinery, which includes a high Rubisco activation state and stromal fructose-1,6-bisphosphatase under high light and low temperatures [41,42]. At 0 °C, both species can maintain around 30% of their maximum photosynthetic rate, which occurs close to their optimum growth temperature, mainly between 10 °C and 19 °C (see [5]). Despite the several studies on the effect of temperature on the photosynthesis of both Antarctic species, there were few aimed at elucidating the mechanisms involved in photosynthetic regulation, i.e., the anatomical and biochemical limitations and its responses to different environmental conditions. In a study evaluating plants growing on King George (62° S) and Lagotellerie (67.5° S) Islands, Sáez et al. [7] determined that regardless of species, population, and temperature measurements (10 and 15 °C), *in situ* carbon assimilation is strongly limited by CO_2_ diffusion within the leaf mesophyll (g_m_). Specifically, g_m_ estimated from gas exchange and chlorophyll fluorescence and from modeled leaf anatomical parameters was remarkably low, restricting CO_2_ diffusion inside the leaf and imposing the strongest restriction for the acquisition of this gas. At the ultrastructural level, some chloroplast traits (size and arrangement) and mainly the cell wall thickness and the cytoplasmic and stromal components determined the strong resistance to CO_2_ mobility through the leaf that was found for both species. Interestingly, ref. [40] reported significant correlations between cell wall metabolism and g_m_ in *D. antarctica*. Several cell wall components (glucuronic and galacturonic acids, fucose, arabins, and *p*-coumaric acid) were altered at low temperatures, suggesting important adjustments in cell wall dynamics. In addition, increased gamma-aminobutyric acid (GABA), which has been suggested as a signaling molecule that inhibits cell expansion and, therefore, regulates growth, was also related to the physiological performance of *D. antarctica*. This reinforces the idea that cell wall composition would affect the CO_2_ pathway through the apoplast to the carboxylation sites and, finally, the photosynthetic performance. The low g_m_ values determined under field conditions [7,17] are similar to those determined under laboratory conditions at different growth temperatures and soil water availabilities [43]. As a compensatory mechanism to the low g_m_, the enzyme ribulose-1,5-bisphosphate carboxylase/oxygenase (Rubisco) of both Antarctic species shows a high specificity for CO_2_ (S_c/o_) and relatively high values of catalytic turnover rates (k_cat_^c^), which suggests a close coordination between CO_2_ diffusion and foliar biochemistry, which may ultimately be essential for optimizing carbon balance in these plant species [7]. In a recent study, Gago et al. [44] evaluated photosynthesis and stress tolerance of *D. antarctica* growing at three sites with contrasting soil nutrient availability. Overall, they found that plants from all three sites showed similar photosynthetic rates, but mesophyll and biochemical determinants were more limiting (~25%) in plants growing on soils with low nutrient availability (where N and P levels are considerably low). Additionally, these plants allocated carbon to metabolites involved in membrane, protein, and cell wall stabilization to prevent the collapse of the structure under higher levels of oxidative stress. These findings suggest that these metabolic rearrangements may contribute to the observed increase in mesophyll and biochemical limitations. This response appears to be an attempt to maintain cell integrity at significant levels of oxidative stress when antioxidant metabolism (which is nutrient-expensive) is constrained by scarce resources. In contrast, when soil nutrients are available, expensive compounds in terms of mineral elements and energy requirements (such as amino acids, secondary metabolites, and polyamines) increase to keep oxidative stress under control. These findings demonstrate that *D. antarctica* exhibits differential physiological performance to cope with adverse conditions depending on resource availability (excess or shortage), enabling it to maximize stress tolerance without compromising its photosynthetic capacity. This metabolic flexibility that depends on nutrient availability is another important feature for plant species inhabiting Antarctica, given the spatially patchy nutrient availability in Maritime Antarctica.

For *C. quitensis,* it has been suggested that anatomical traits determining g_m_ and, therefore, net photosynthesis are complemented by a recently studied mechanism called alarm photosynthesis [45]. This mechanism appears to act as a biochemical process that assimilates CO_2_ derived from the decomposition of calcium oxalate (CaOx) crystals. In this context, Gómez-Espinoza et al. [45] found that *C. quitensis* exposed to CO_2_ limitation (11.5 ppm) exhibited a significant increase in CaOx crystal decomposition and oxalate oxidase activity in their leaves. This suggests that under stress conditions, crystal decomposition could potentially provide CO_2_ molecules to the mesophyll tissue. In parallel, the electron transport rate (ETR) slightly decreased but remained stable when compared to the control group. Additionally, under nonphotorespiratory conditions, significant CaOx crystal decomposition was also observed, whereas the ETR decreased by approximately 40%, although it remained adequate for maintaining a baseline level of photosynthesis if required. Despite having neither atmospheric CO_2_ nor O_2_ as final electron acceptors, *C. quitensis* plants were able to maintain a stable ETR value of around 20 μmol e m^−2^ s^−1^ for about 10 h of stress. It is possible that these ETR values were derived from the use of CO_2_ supplied by the CaOx crystals as final electron acceptors. Therefore, for *C. quitensis*, alarm photosynthesis could play a significant role as a complementary endogenous mechanism that would facilitate carbon supply in response to the limitations in CO_2_ diffusion associated with its anatomical features, which are necessary for its tolerance to the extreme Antarctic climate.

In relation to the uniqueness of the Antarctic vascular plants, Clemente-Moreno et al. [39,40] evaluated the effects of low temperature on several metabolites associated with photosynthesis in plants grown at different temperatures (4, 15, and 23 °C), where the Antarctic species were compared with some non-Antarctic species belonging to the same family as the Antarctic species (*Triticum aestivum* L., TA and *Dianthus chinensis,* DC). Both Antarctic species showed different strategies for coping with low temperatures than the two phylogenetically related non-Antarctic species. In particular, the metabolite trait-dependent association of *D. antarctica* included various sugars, secondary metabolite precursors, and photosynthesis-related cell wall components. In contrast, the non-Antarctic species TA showed associations between some sugars and secondary metabolite precursors with A_N_. The results showed glucose and trehalose accumulations and reduced levels of secondary metabolites in TA when A_N_ declines. Particularly strong negative associations were observed between A_N_ and pyruvate, benzoate, proline, methionine, glucuronate, and fumarate in *D. antarctica*, suggesting lower carbon and energy investment in central metabolic pathways, such as glycolysis and the tricarboxylic acid cycle [40]. In *C. quitensis*, the metabolite trait-dependent associations showed strong and positive associations between A_N_ and glycine, mannitol, Unknown-03, and Unknown-06. In contrast, in the non-Antarctic species DC, negative relationships were found between A_N_ and glycine and mannitol, indicating a stress response putatively associated with photorespiration and polyol metabolism. These authors suggest the activation of metabolic pathways related to polyamines, secondary metabolism, and aspartate in *C. quitensis*, which were negatively correlated with A_N_, opposite to the response observed in DC. Likewise, strong and negative relationships in *C. quitensis* between A_N_ and ascorbate, methionine, and Unknown-08 would indicate the activation of sulfur (S) metabolism and the increase in antioxidant capacity when photosynthesis falls [39].

### 2.3. Respiration

Leaf respiration plays a key role in determining plant growth and survival, but it also has a strong influence on the net CO_2_ exchange in ecosystems and atmospheric concentrations of this gas [46,47]. Respiratory energy (ATP and reducing equivalents) is necessary for cell maintenance and growth and provides the carbon skeletons necessary for the synthesis of cell and tissue components [48,49,50,51]. However, despite its importance, very little has been addressed on the respiratory metabolism of the Antarctic vascular species. In [52], Edward and Smith reported similar respiration rates between both species below 10 °C, which were notably lower at temperatures near 0 °C. An increase in respiration rates in plants grown at low temperatures is widely accepted as cold acclimation [53,54]. Interestingly, Clemente-Moreno et al. [39] found that respiration rates were not affected by growth temperature (4, 15, and 23 °C) in *C. quitensis*, indicating an important robustness and acclimation of this parameter to low temperature, coinciding with that previously proposed by Xiong et al. [55]. 

Under oxidative stress, mitochondrial respiration has been suggested to modulate reactive oxygen species (ROS) formation by dissipating reductants from both mitochondria and chloroplasts [56,57]. Plant mitochondria have two electron transport pathways: the cytochrome oxidase (COX) pathway, where electrons from the ubiquinone (UQ) pool are transferred to COX, and the alternative oxidase (AOX) pathway, in which electrons are transferred from the UQ pool to AOX, decreasing the energy efficiency of respiration because it bypasses two of the three sites of proton extrusion in the mitochondrial electron transport chain (mETC) [54,58]. High levels of carbohydrates in leaves have also been associated with high AOX activity in vivo because they play an important role as a “salvage” pathway against excess electrons, efficiently consuming the reducing equivalents, preventing over-reduction of the electron chain, and avoiding ROS production [59]. Therefore, changes in the electron transport pathways could also modulate cellular oxidative stress and energy levels in Antarctic vascular plants, which, under both field and controlled conditions, present high levels of nonstructural carbohydrates [19,21,28,30], suggesting a high availability of substrates for respiration [29,60]. Given that in *C. quitensis,* low temperatures reduce photosynthesis (up to 88%), which would indicate that it is not their main strategy to avoid ROS formation [39], it is possible to infer that respiration is the main process helping to avoid oxidative stress. In fact, remarkably high respiratory rates have been reported in *C. quitensis*, between 3.0–4.6 µmol CO_2_ m^−2^ s^−1^ [7,39,41]. Thus, it is likely that the COX pathway plays a relevant role in *C. quitensis* since the maintenance of high respiratory rates through the cytochrome pathway would ensure the production of mitochondrial ATP at low temperatures for growth and invest in the biosynthesis of sulfur compounds and polyamines, which play important antioxidant roles [39]. In addition, Sanhueza et al. [60] evaluated the diurnal and nocturnal variations in the respiration rates of both Antarctic species and suggested that in *C. quitensis*, a significant number of soluble sugars are mobilized from the leaves to the roots, mainly during the night, which also suggests the occurrence of high respiration rates at the root level. In contrast, the extraordinary capacity of *D. antarctica* to maintain respiration at low temperatures [61] is probably linked with the AOX pathway. Clemente-Moreno et al. [40] postulate that the main strategy displayed by *D. antarctica* to cope with low temperatures in the field is a highly organized leaf senescence program to re-assimilate nitrogen. This increased nutrient mobilization provides higher levels of carbon skeletons to the tricarboxylic acid cycle (TCA cycle) [62], increasing the activity of the mETC; hence, an active AOX would be essential to avoid overreduction. In addition, Sanhueza et al. [63] suggested that respiratory acclimation in *D. antarctica* was related to the plasticity of mitochondria structure, involving changes in abundance and structure, and reflecting the production of new respiratory components, mainly at the mETC level. Overall, these results suggest that both Antarctic species seem to deploy different respiration metabolism at low temperatures. However, the dynamics of carbohydrate allocation, the role of carbohydrates in root respiration, and the role of the COX and AOX pathways are still unknown. 

### 2.4. Photoprotective Mechanisms

The combination of stressors such as low temperature and high irradiance, usually found during the Antarctic growing season, favors photoinhibition [64,65,66]. However, both Antarctic vascular species have shown a high capacity to cope with these conditions, at least in the laboratory [65,67], although they display slightly different photoprotective strategies. According to Pérez-Torres et al. [67,68], *D. antarctica* grown at low temperatures actively uses oxygen as an alternative electron sink through the water-water cycle to protect PSII from excessive light, coordinated with a robust antioxidant machinery, with a particularly high SOD and APX activity. However, Clemente-Moreno et al. [40], when they evaluated the antioxidant response at low temperatures of the main ROS sequestering enzymes (APX, POX, and CAT), did not observe significant increases in these enzymes. In contrast, *D. antarctica* displayed a rather unique pattern of antioxidant enzyme activities after prolonged exposure to low temperatures. Additionally, these authors reported positive relationships between POX activity and some physiological traits (A_N_, photorespiration, and the maximal photochemical efficiency of the PSII) but a negative relationship between membrane lipid peroxidation and POX activity [40]. *C. quitensis* regulates its electron transport pathway by preventing O_2_ reduction. Pérez-Torres et al. [69] reported lower levels of antioxidants compared to other plant species, suggesting a low contribution of the water-water cycle to the modulation of the redox state of the photosynthetic electron transport chain. In contrast, Clemente-Moreno et al. [39] demonstrated that *C. quitensis* can display a constitutively high antioxidant capacity related mainly to sulfur-containing metabolites and other secondary shikimate metabolism, with no major variation in its antioxidant enzymatic activities. Additionally, there is evidence that cold exposure of *C. quitensis* from Antarctica may stabilize the electron transport chain polypeptides within the thylakoid membrane or maintain their turnover. This was associated with a fast recovery from low temperature-induced photoinhibition [70].

Both Antarctic species have shown increases in the thermal dissipation of excess light energy at low temperatures and high irradiance. However, only *C. quitensis* exhibits substantial increases in the thermal dissipation capacity (NPQ_max_) through activation of the xanthophyll cycle [65,67,69]. Recently, Sáez et al. [71] reported that *D. antarctica* maintains a xanthophyll pool slightly de-epoxidated at dark when grown at low temperatures, suggesting that this species is prepared to respond to high irradiance during cold mornings or after severe nocturnal frosts, which could constitute another mechanism of this species to cope with harsh Antarctic conditions. Additionally, these authors reported a lack of correlation between NPQ, DEPS (de-epoxidation state of the xanthophyll cycle pigment pool), and delta de-epoxidation (ΔDEPS), indicating two possible explanations. First, *D. antarctica* may have a zeaxanthin-independent qE (energetic quenching), or alternatively, qT (transition state-related) may play a more significant role in the heat dissipation of this species. It seems that the adaptative traits for coping with excess radiation at low temperatures have been developed to avoid excess light capture, i.e., preserving the photochemical apparatus rather than in the direction of enhancing dissipation mechanisms through nonphotochemical processes. This capacity to preserve the photosynthetic apparatus is manifested in the field through a high maximal quantum yield of PSII (F_v_/F_m_) [72], which is maintained under laboratory conditions [71]. Regarding this, at low temperatures, *D. antarctica* displays a lower qL (photochemical quenching) related to a reduced antenna size (higher Chl a/b and β-carotene/neoxanthin ratio). Therefore, this could be a useful strategy to avoid an excess of absorbed light and to maximize the photochemical use of this energy. Altogether, the evidence supports the idea that *D. antarctica* has efficient mechanisms to avoid excess light energy absorbed in the low temperatures of Antarctica.

In C_3_ plants, it has been shown that photorespiration constitutes an important alternative electron sink under stress conditions, especially when CO_2_ availability decreases [73]. Although photorespiration may not be a mechanism that prevails in Antarctic vascular species, especially due to the high Rubisco specificity factor [7], it is likely that photorespiration plays a role as a safe alternative pathway for electron transport when the availability of CO_2_ at the carboxylation site is limited. Thus, in *C. quitensis*, the low photosynthetic rates at low atmospheric CO_2_ concentrations due to low g_m_ are consistent with high values of the relationship between the rate of electron transport and the gross rate of CO_2_ assimilation (ETR/A_G_); this is indicative of increased photorespiration rates. Additionally, the presence of many organelles (mitochondria or peroxisomes) around chloroplasts in both Antarctic species has been suggested as facilitators for the exchange of CO_2_ between the respiration and photorespiration processes, a phenomenon already postulated by Gielwanoeska and Szczuka in [74]. Clemente-Moreno et al. [39,40] reported a significant decrease in photorespiration in both species at low temperatures (4 °C). In *C. quitensis*, it was suggested that when photosynthesis and photorespiration are restricted at low temperatures, N (nitrogen) and S metabolism could play a key role in cellular homeostasis by consuming ATP, reducing equivalents (NADPH), and skeletons of carbon produced by the TCA cycle.

Finally, it has been observed that *D. antarctica* growing in Antarctica frequently shows a mixture of green and yellow leaves, which could be considered a symptom of leaf senescence. During this process, chloroplasts are dismantled in a highly organized manner to preserve cell function for as long as possible [75]. Decommissioning is a complex process that can proceed through different pathways. The renewal of chloroplast components provides nutrients for sink tissues and may be relevant to controlling the senescence progression [75]. This could be associated with those reported by Clemente-Moreno et al. [40] regarding the senescence program activated by *D. antarctica* to dismantle major cellular structures.

## 3. How Unique Are the Antarctic Plants in the Antarctic Environments? What Is Known about the Performance of *C. quitensis* and *D. antarctica* at Different Locations?

Although some information has accumulated on the physiological mechanisms that these species deploy in different populations within Antarctica [7,40,71,76], ecophysiological studies outside Antarctica are sparse for *C. quitensis* but nonexistent for *D. antarctica.*


Given the wide geographic distribution of *C. quitensis* and the isolation in which their populations have been developed, ecotypic differentiations in different ecophysiological traits have been reported [19,65,66,77]. Moore [9] studied the morphological variations in *C. quitensis* populations throughout its distribution and described a wide variability in the characteristics related to leaf length and the leaf length/width ratio, as well as the characteristics related to floral morphology. In particular, they detected that the greatest variations were found in the extreme south of the distribution (southern Patagonia, 50° S; Tierra del Fuego, 54° S; and the Falkland Islands, 51° S), suggesting that these variations could be related to adaptations controlled by the environment, such as water availability, degree of exposure, and anthropogenic pressure; however, they could also have a genetic basis. Gianoli et al. [19], when comparing populations of La Parva (33° S) in the Andes of central Chile and Antarctica under controlled common garden conditions, reported through studies of freezing resistance, morphology, and sequences of the internal transcribed spacer region (ITS) the existence of an ecotypic differentiation between populations. Likewise, Bascuñán-Godoy et al. [77] described differences in leaf anatomy between the Andean and Antarctic ecotypes. In addition, Cavieres et al. [5] showed a clinal pattern of variation in the microstructural features among higher and lower latitudes (Punta Arenas, 58° S; King George Island, 62° S; and Lagotellerie Island, 67.5° S). In particular, towards higher latitudes, there are thicker and cylindrical leaves, evidenced by a lower cross-sectional area, thicker mesophyll, narrow adaxial surface, and reduced epidermal thickness [5,78]. Recently, Gómez-Espinoza et al. [79] reported morphological variations in *C. quitensis* plants from a latitudinal gradient (La Parva, 33° S; Punta Arenas, 53° S; and Antarctica, 62°), such as habit size (i.e., plant height, width, and overall visual appearance) and differences in leaf length and width. Plants from the Antarctic population had the smallest shoots and length of leaves. Additionally, these authors reported the presence of CaOx crystals in all *C. quitensis* provenances, demonstrating that the greatest abundance and size of these crystals were found in Andean provenances. It has been suggested that these variations in the foliar morphoanatomical characteristics seem to be a constitutive adaptation [19,65,78,80] since the differences between the traits are partially maintained in the common garden, indicating that these functional adaptations could have a genetic basis. Therefore, they can be attributed to continuous selection processes, where several characteristics can vary in response to the environmental conditions that prevail in each specific habitat [80].

Regarding gas exchange, both under field and controlled conditions, the Antarctic ecotype of *C. quitensis* presents higher net photosynthesis values (A_max_) than the Andean ecotype [39,65,66]. Under field conditions, Antarctic individuals show a photosynthesis range of 8 to 11 µmol CO_2_ m^−2^ s^−1^, while photosynthetic rates around 5 µmol CO_2_ m^−2^ s^−1^ have been reported in the Andes [7,41,64]. Sierra-Almeida et al. [66], when evaluating the optimal photosynthetic temperature (T_op_) in *C. quitensis* from Antarctica and the Andes of central Chile under controlled conditions (4 and 15 °C), showed that both the growth temperature and population had significant effects on T_op_. At low temperatures, T_op_ changed in both provenances, which suggests some acclimation capacity. Similarly, both the compensation and saturation points are higher in the Andean population, suggesting marked differences in light requirements [65,66]. Thus, when evaluating the light response curves from the Andes (La Parva) and Antarctica at all the intensities of photosynthetically active radiation (PAR) applied, the Antarctic population presents the highest electron transport rates [79]. Therefore, the highest photosynthesis and the lowest compensation and saturation points at low temperatures demonstrate the ability of the Antarctic ecotype to maximize its photosynthetic performance, thus optimizing the energy allocated for growth and reproduction in a short period with favorable temperatures and very unstable light supply for photosynthesis [65,66].

Although photoinhibitory conditions can occur in both locations (Antarctica and Andes), they differ in the degree and extent of this stress [64,65,77]. Bascuñán-Godoy et al. [76], when evaluating both *C. quitensis* ecotypes under their respective natural conditions, reported that both are highly efficient in maintaining their photochemical processes coordinated with increases in thermal dissipation of excess energy (NPQ), specifically associated with the fast relaxation component (NPQ_f_) and the de-epoxidation state of the xanthophyll pool. Notwithstanding, these ecotypes exhibit different photoprotective strategies [77]. While the Antarctic ecotype shows higher levels of qL and a fast-relaxing component of nonphotochemical quenching (NPQ_f_), the Andean ecotype exhibited an increase in the slow relaxation component (NPQs), suggesting a greater sensitivity to photoinhibition induced by low temperatures [65,77]. Therefore, while the Andean ecotype is prepared to face high irradiances at relatively high temperatures, the Antarctic ecotype is better able to resist the combination of low temperatures and high irradiances [65].

Regarding *D. antarctica,* Moore [9] reported that this species varied sharply in some leaf morphological characteristics, such as leaf width and degree of folding and ligule length over a latitudinal gradient. In general, there is a trend for populations of the South Orkney Islands, South Shetland Islands, and the Antarctic Peninsula to have shorter ligules and narrower, more folded leaves than the Tierra del Fuego (54° S) and Patagonian (50° S) populations. Similarly, differences in leaf anatomical and chloroplast traits have been reported for *D. antarctica* growing throughout Maritime Antarctica [74,81]. Despite these leaf anatomical differences, Edwards and Smith [52] reported no differences in the temperature response curve of photosynthesis in *D. antarctica* collected from different locations (South Georgia (54° S), South Orkney Islands (60° S), and Margarita Bay (67° S)). Likewise, Sáez et al. [7] did not find differences in leaf anatomy between plants from King George and Lagotellerie Islands, except for cell wall thickness, which was greater in the southern Lagotellerie Is., and the surface area of chloroplasts compared to intercellular air spaces per leaf area (S_c_/S), which was higher on King George Island.

Therefore, although *C. quitensis* and *D. antarctica* can grow throughout the Andes, there is still a lack of knowledge regarding the physiological mechanisms (i.e., photosynthesis, photoprotection, respiration, etc.) that these species deploy at different populations outside Antarctica. It would be interesting to take advantage of the wide distribution of Antarctic vascular species and use them as model organisms to elucidate the extremophile plant responses to changes in environmental conditions without having phylogenetic noise. This effort would contribute significantly to understanding the possible responses and plant adaptations to climate change.

## 4. Consequences of Warming on the Ecophysiology of Antarctic Vascular Plants

The warming trend in the Antarctic Peninsula has been reported as the second fastest on Earth during the last century, with sustained increases in extreme high temperatures and reaching historical records [82]. Paradoxically, regional warming that promotes the growth and reproduction of Antarctic plant species [13] could reduce their ability to survive, making even the best cold-adapted plants more susceptible to damage from freezing temperatures.

*In situ* warming experiments using open-top chambers (OTC) as passive and continuous warming systems have shown that after two seasons of warming, the LT_50_ slightly increased in two out of the three studied sites for *C. quitensis* and in one of the three studied sites for *D. antarctica* [18]. These authors reported that Antarctic plants could be frequently exposed to temperatures ranging from ~7.8 °C to −13 °C during the summer–autumn transition, respectively, suggesting that, under this scenario, *D. antarctica* has a temperature safety margin of 10 °C since the plants exposed to warming generally maintain their LT_50_ about −23 °C. However, this safety margin would not exist for *C. quitensis* due to a significant increase in LT_50_. Although both species can cope with summer frost events, if environmental temperatures continue to increase, as some authors propose [83], they could dangerously approach the range where *C. quitensis* plants may experience frost damage.

In addition, it was determined that the increase in temperature around 3 °C induced by the OTCs triggered leaf anatomical modifications that resulted in significant changes in g_m_ [17]. However, these modifications were only appreciable in *C. quitensis*. In *D. antarctica*, most of the anatomical features associated with CO_2_ mobility were not altered by the OTC (see Figure 1). For *C. quitensis*, the leaf anatomical changes that induce a greater g_m_ resulted in increases in photosynthetic assimilation, thus promoting greater carbon gain and plant growth. In this species, these changes were accompanied by alterations in the chemical composition of the leaf (lower fiber content), concomitantly with lower dry mass per leaf area (LMA) and leaf density (LD), reflecting a structural control over the mesophilic diffusion limitations for photosynthesis. This, together with the greater proximity of the chloroplasts to the cell wall, less thickness of chloroplasts, and greater proportion of the mesophyll surface exposed to the air spaces, constitutes an important anatomical factor that results in an improvement of the internal transfer of CO_2_ under *in situ* warming conditions [17]. Under controlled laboratory conditions, it was confirmed that warmer conditions favor the photosynthetic capacity of *C. quitensis* and *D. antarctica*, although the latter requires higher temperature increases to show the same response (see Figure 1). Additionally, it was evidenced that the moderate water deficit can completely counteract any benefit for photosynthesis induced by the increase in temperature, which suggests that these species may present a homeostatic photosynthetic response to climate change expected for the Antarctic region [43].

Recently, López et al. [23] evaluated the effect of daytime and nighttime warming on freezing resistance and cryoprotectant accumulation of Antarctic vascular plants under laboratory conditions. The results revealed that night warming induced a reduction in the freezing resistance in *D. antarctica* and *C. quitensis*. Overnight warming led to an increase in the LT_50_. However, in *D. antarctica*, diurnal warming per se was not able to induce deacclimatization to cold. This discrepancy between laboratory and field experiments indicates the possible influence of other factors within the OTC. For example, the absence of nocturnal warming inside OTC can exert a substantial influence on freeze/thaw events in the soil and root system and, consequently, on carbon and nitrogen decomposition processes in the soil. Additionally, these authors reported that both species reduce their sucrose content by more than 28% with warming. However, sucrose seems to have a more relevant role in freezing resistance as a cell osmoprotectant in *C. quitensis* since a drop of 30% in sucrose content could be related to a reduction in resistance to freezing of its leaf tissue when the growth temperature increases. Likewise, a decrease in the raffinose content was reported with warming in *C. quitensis*. Thus, raffinose could also be particularly important in the response to resistance to freezing because the LT_50_ method used by these authors is essentially based on the stability of PSII. On the other hand, night warming reduced the expression of dehydrin-like peptides in *D. antarctica*, while similar levels of proteins were reported in cold-acclimated plants and with daytime warming per se. This suggests that dehydrins play a key role in freezing tolerance in this species and further supports the idea that nocturnal rather than diurnal warming favors the cold deacclimatization response in *D. antarctica.* Since plants should not express their cryoprotective mechanisms during cold deacclimatization, CBF genes are expected to be downregulated. When the effect of daytime and nocturnal warming on the genetic expression of *D. antarctica* under laboratory conditions was evaluated, it was determined that only nocturnal warming can downregulate the genes related to freezing tolerance. On the contrary, nocturnal warming induced genes related to growth promotion and carbon assimilation, which induce cold deacclimatization in *D. antarctica* [38]. These results further support the idea that nocturnal rather than diurnal warming favors the deacclimatization response to cold in *D. antarctica*. However, further studies are needed to understand the consequences of regional warming on field populations of *D. antarctica*. In this sense, it would be interesting to evaluate how gene expression is affected under night warming in experimental field conditions.

Regional warming models for the Antarctic Peninsula predict a higher frequency of extreme events [83]. Therefore, extreme warming events can distort the energy balance and trigger a biochemical limitation of photosynthesis. Furthermore, increases in temperature can have significant effects on the partition of absorbed light energy towards photochemical conversion and thermal dissipation. Several studies have shown that cold acclimation induces some of the previously mentioned photoprotective mechanisms in Antarctic vascular plants [65,67,77]. Cold acclimation also favors the recovery of photoinhibition, probably by inducing repair mechanisms [70]. Therefore, long-term warming may limit the acquisition of fully functional photoprotective mechanisms such as those mentioned above, which, in turn, may result in a reduction in plant performance. In this sense, warming studies under laboratory conditions suggest that a moderate increase in temperature favors photochemical activity but reduces thermal dissipation responses (NPQ and xanthophyll cycle) [71]. This could, in the long term, cause a decrease in photoprotective mechanisms and leave plants more vulnerable to photoinhibition induced by extreme stochastic daytime frost events.

While increased temperatures on the Antarctic Peninsula could have a dramatic effect on both Antarctic vascular plants, a thermal acclimation of respiration could reduce carbon loss generated by warmer temperatures. This would allow maintaining the basal respiratory rate, which would contribute to improving the net carbon assimilation. In recent studies focused on the respiratory responses to experimental warming and its incidence on the carbon balance, different respiration responses were reported between the Antarctic species. Whilst diurnal warmer conditions increase R_d_ in *C. quitensis*, no changes occurred in *D. antarctica* [39,40]. Nocturnal warming conditions improved the carbon balance of both Antarctic species through different mechanisms involving respiratory acclimatization in *C. quitensis* and increased carbon uptake in *D. antarctica* [60].

Along with the increase in temperature, changes in atmospheric CO_2_ concentrations have also been reported within Antarctica [84,85]. The impact of warming and/or elevated CO_2_ on carbon metabolism will depend on their differential effects on photosynthesis and respiration. Recently, Sanhueza et al. [63] evaluated the effects of elevated CO_2_ concentrations and nighttime warming on gas exchange, nonstructural carbohydrates (total soluble sugars and starch), respiration-related enzymes, and mitochondrial traits (number and sizes of mitochondria) in both Antarctic species (see Figure 1). In general, *C. quitensis* and *D. antarctica* displayed different acclimation mechanisms to the combination of elevated CO_2_ and nocturnal warming. In *C. quitensis*, a reduced ability to maintain photosynthetic performance and a lack of respiratory response (demonstrated by a low short-term sensitivity in relative protein abundance and mitochondrial traits) induce a significant increase in the photosynthesis and respiration ratio, suggesting possible damage to the foliar carbon balance in this species. Thus, in *C. quitensis*, most of the physiological parameters evaluated suggest a low capacity for respiration acclimatization at high CO_2_ and nocturnal warming. On the contrary, in *D. antarctica*, the ability to maintain high photosynthetic rates on warm nights and high CO_2_ seems to be related to the ability to modify traits related to mitochondrial structure (a reduction in the number of mitochondria and an increase in mitochondrial size), indicating a high level of respiration acclimation to warming, which contributes to maintaining the foliar carbon balance. Thus, the high capacity for morphological and physiological adjustments of *D. antarctica* seems to be an important trait that helps it tolerate environmental changes and could contribute to increasing its ability to colonize and expand successfully throughout the Antarctic Peninsula.

## 5. Conclusions and Future Perspectives

Although it has been established that the foliar xeromorphic anatomical characteristics (LD and LMA) of both Antarctic species are key to withstand the harsh climatic conditions of Antarctica, recent studies highlighted the importance of cell wall (thickness and composition) as a key trait related to water loss control and CO_2_ transference in the leaf, therefore controlling the leaf gas exchange of Antarctic vascular species, regardless of the growing conditions. This adaptation raised new research areas, including the relevance of Rubisco kinetics traits, crucial for maintaining positive carbon gain in the harsh Antarctic environment and the metabolomic routes that explain gas exchange, nutrient uptake, and photoprotective mechanisms (to summarize the response shown in Table 2). This new knowledge has contributed to understanding the adaptations that allow Antarctic plants to grow and develop under eminent harsh environmental conditions.

Additionally, recent studies show that climate change can have significant and contrasting effects on the morphophysiological processes of Antarctic species, with *C. quitensis* responding rapidly to warmer conditions and *D. antarctica* being less responsive, particularly under warmer field conditions (see Figure 1). These results open a paradox about the positive (more photosynthesis and growth, population expansion) and negative (reducing freezing tolerance) effects of warming. In addition, the lack of studies in different locations of Antarctic plants is evident, especially outside Antarctica. Using the latitudinal gradient as a natural laboratory to predict the effects of climate change could help us to predict the real effects of this change on Antarctic species and others that share the gradient. Thus, this updated knowledge serves to formulate new questions, among them: What factors promote/restrict the distribution of Antarctic species at different spatial scales? Are there other adaptative traits that allow Antarctic plants to cope with harsh climatic conditions? i.e., hydraulic properties, NSC store-allocation patterns, specific genes, etc. How does regional warming affect the photoprotection of Antarctic species? Do Antarctic plants, especially *D. antarctica*, develop similar morphophysiological traits to deal with the harsh environment when growing outside of Antarctica? And how will they respond to warming? And finally, are Antarctic plant responses to warming time-dependent? These questions must be answered to elucidate how unique the Antarctic species are and to know whether their adaptative mechanisms will be an advantage or disadvantage in responding to changes in environmental conditions.

## Figures and Tables

**Figure 1 plants-13-00449-f001:**
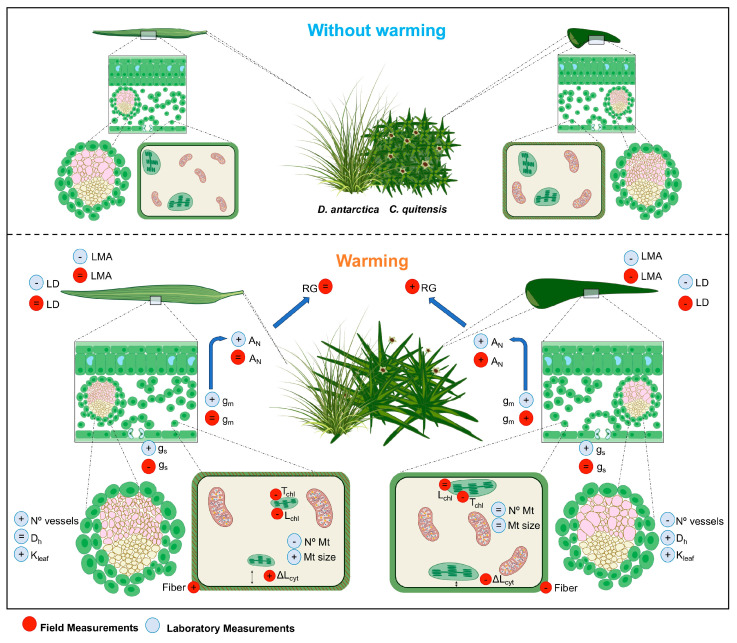
Morphophysiological traits and their responses to warming in *D. antarctica* and *C. quitensis* under field (red circles) and laboratory (blue circles) conditions. RG, relative growth; A_N_, net CO_2_ assimilation rate; LMA, leaf mass area; LD, leaf density; g_s_, stomatal conductance; g_m_, mesophyll conductance to CO_2_; T_chl_, chloroplast thickness; L_chl_, chloroplast length; ΔL_cyt_, distance from the chloroplast to the cell wall; D_h_, hydraulic diameter; K_leaf_, leaf hydraulic conductivity; Mt; mitochondria. Minus, plus, and equal signs inside the red and blue circles indicate a decrease, increase, and unchanged in the described traits compared to the condition without warming (figure created using BioRender.com).

**Table 1 plants-13-00449-t001:** Freezing resistance of Antarctic vascular plants. Values correspond to mean LT_50_, determined for non-acclimated (NA) and cold-acclimated plants (CA) after exposure to temperatures ranging from 2 to 5 °C for a period of 21 days, except for Chew et al. [20], where the cold acclimation period was 14 days.

	*Deschampsia antarctica*		*Colobanthus quitensis*		Freezing Injury Method
Reference	NA	CA	NA	CA	
Bravo et al. [21]	−12.0	−26.6	−4.8	−5.8	Ion leakage
Gianoli et al. [19]	−	−	−7.0	−15.0	Plant survival
Chew et al. [20]	−12.0	−17.0	−	−	Survival and regrowth
Reyes-Bahamonde [22]	−16.5	−18.4	−7.0	−14.9	Photoinactivation
López et al. [23]	−16.9	−24.4	−8.1	−13.3	Photoinactivation

**Table 2 plants-13-00449-t002:** Summary of the mechanisms that underlie the ability of both Antarctic vascular plants to resist environmental stresses.

Mechanisms	*Deschampsia antarctica*	*Colobanthus quitensis*
Freezing Resistance	- Freezing tolerance. - High activity of antifreeze proteins in the apoplast. - High dehydrin activity. - Highly unsaturated phosphatidylglycerol (PG) fraction. - High sucrose concentration associated with high SPS activity. - Enrichment in genes related to cold tolerance. - Accumulation of metabolites related to osmoprotection and membrane stabilizers.	- Freezing tolerance - High sucrose concentration associated with high SPS activity. - Accumulation of metabolites related to osmoprotection and membrane stabilizers.
Photosynthesis	- Compensatory mechanism between low g_m_ and high Rubisco specificity factor for CO_2_. - Associations between cell wall metabolites and photosynthesis. - Strong associations between several metabolites with photosynthesis.	- Compensatory mechanism between low g_m_ and high Rubisco specificity factor for CO_2_. - Alarm photosynthesis: use of CaOx as a source of CO_2_. - Strong associations between several metabolites with photosynthesis.
Respiration	- Increased AOX activity.	- -High respiratory rates, through the COX pathway, associated with the biosynthesis of sulfur compounds and polyamines.
Photoprotection	- Water-water cycle associated with a robust photosynthetic machinery. - Unusual POX activity. - Maintain a xanthophyll pool slightly de-epoxidated at dark when grown at low temperatures. - Remobilization of nutrients.	- Substantial increases in its thermal dissipation capacity (NPQ_max_) by activating the xanthophyll cycle. - High antioxidant activity associated with sulfur metabolism and secondary metabolism, with normal antioxidant activity. - Photorespiration as an alternative route.

## Data Availability

Data will be shared on request to the corresponding author.

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
