# Peer review of "Ecophysiology of Antarctic Vascular Plants: An Update on the Extreme Environment Resistance Mechanisms and Their Importance in Facing Climate Change"

_plants, 2024, doi:10.3390/plants13030449_

Round 1
Reviewer 1 Report
Comments and Suggestions for Authors
In this manuscript, the authors reviewed research progress of ecophysiology of antartic vascular plants, it is of good help for understanding plants’ tolerance to extreme low temperature and the possible mechanism of plants’ response to long-term low temperature. The manuscript is well organized, understandable and interesting. The review is enlightening for research about plant stress resistance, as well as ecological response of plant to global warming.
Author Response
Reviewer 1: Comments for the Author
R1: We appreciate the reviewer's comments.
Reviewer 2 Report
Comments and Suggestions for Authors
The manuscript entitled Ecophysiology of Antarctic Vascular Plants: An Update on the Extreme Environment Resistance Mechanisms and their Importance Facing Climate Change by Ramírez et al. is a well written manuscript. This interesting review of Antarctic vascular plants that includes different ecophysiological aspects like freezing resistance, robust carbon assimilation, respiration rates and photoprotective mechanisms. Also, there is an interesting discussion related to differences in some physiological traits between the different populations that are found in locations with different environmental conditions. Finally, the effects of warming on the ecophysiology of the antartic plants was presented and discussed. In order to have a complete approach to these singular plants, the plant cycle of the two plants Colobanthus quitensis and Deschampsia antarctica must be included. Also, if there are differences in plant cycle reported between populations could be important to understand the relevance of these plants.
Author Response
Reviewer 2. Comments for the Author
R2. In order to have a complete approach to these singular plants, the plant cycle of the two plants Colobanthus quitensis and Deschampsia antarctica must be included. Also, if there are differences in plant cycle reported between populations could be important to understand the relevance of these plants.
Response: The singularities of these two angiosperms are really focus on their morphophysiological traits which allows their performance under these harsh environmental conditions. However, there are no reports that evidenced that these two species, one from Caryophyllacea family and another Poaceae family have distinctive features in their life cycle compared to other Caryophyllacea or Poaceae, respectively. Therefore, this is information already described for angiosperms in Botany textbooks.
Reviewer 3 Report
Comments and Suggestions for Authors
The plants-2641000 reviewed ecophysiological resistance mechanisms of the two Antarctic vascular plants, Colobanthus quitensis and Deschampsia antarctica. This is a comprehensive review, which describes ecophygiological strategy of these plants to cope with Antarctic environments and future climatic change.
Some part of the MS seems to be just a row of paper description with only two tables and one figure. The authors need to revise the MS to be more ready to read.
I recommend adding some tables or figures to summarize the strategy of C. quitensis and D. antarctica for each section such as photosynthesis, respiration, and photosptotective mechanisms. I recommend adding a table to summarize the difference (or similarity) between Antarctic plants and plants distributed in high altitude.
There are other points that need to be clarified in the MS. Followings are the specific comments.
1. Line 25. What is “this knowledge”?
2. Line 62. At the same time?
3. Line 64-35. The meaning of “biological point of view” is unclear. In the above parts of the MS, the features of the Antarctic plants explained at the biological point of view. Please be more specific. Line 24-25 should be also changed.
4. Line 89. The meaning of “different” is unclear.
5. Line 109. Please add an explanation for LT50.
6. Line 112. What is the meaning of “complete individuals without nucleating agents”?
7. Line 117, Table 1. Please add explanation for the condition of cold acclimation.
8. Line 130. Please insert line break before “Different studies”.
9. Line 133. What is the meaning of “wide cellular distribution”?
10. Line 137. What is the meaning of “negative carbon balance”?
11. Line 150-151 and Line 154-160. These parts seem to be inconsistent. The former sentence stated that accumulation of sugar could be caused by reduction in growth. However, the latter sentences explained that high sucrose concentration is related to high activity of sucrose phosphate synthase, which can provide carbohydrate to supply rapid growth.
12. Line 168. What is the meaning of “among them”?
13. Line 184. Under high light?
14. Line 191. What is the meaning of “despite species specific”?
15. Line 207-209. I cannot understand this sentence.
16. Line 213-245. I cannot understand this part. How are these explanations related to adaptation to Antarctic environment?
17. Line 238 – 242. This part is complicated. How is the relationship between ETR and CaOx crystals?
18. Line 250 – 255. I cannot understand the difference between Antarctic species and non-Antarctic species.
19. Line 262. What are “Unknown-03 and Unknown-06”?
20. Line 286-325. How are these explanations related to adaptation to Antarctic environment? The authors stated that Antarctic plants have high capacities to cope with photoinhibition (line 329-330), so oxidative stress may hardly occur.
21. Line 331. Different from what?
22. Line 376. Please use lowercase 3 for C3.
23. Line 376-401. It is better to move this part to the section of “2.2 photosynthesis (line 181)” and rearrange the sentences.
24. Line 427. Thinner cross section => thicker cross section?
25. Line 428. What is the meaning of “narrow abaxial surface”?
26. Line 431. What is the meaning of "habit size”?
27. Line 451-452. Light response curves?
28. Line 486-488. The authors explained physiological mechanisms in the above section in detail. Please be more specific.
29. Line 494-612. This part is highly descriptive and hard to read. Please rearrange the sentences, and use the figure 1 in this part.
30. Line 614-615. Where is the explanation for the xeromorphic anatomical characteristics?
31. Table 2. There is no need using a table. Please describe in the text.
Comments on the Quality of English Language
Moderate English editing is needed.
Author Response
Reviewer 3. Comments for the Author
R3. I recommend adding some tables or figures to summarize the strategy of C. quitensis and D. antarctica for each section such as photosynthesis, respiration, and photoprotective mechanisms. I recommend adding a table to summarize the difference (or similarity) between Antarctic plants and plants distributed in high altitude.
Response: Thank you for this suggestion. We have included a new table (Table 2) which summarizes the physiological suggested strategies of C. quitensis and D. antarctica.
R3. 1. Line 25. What is “this knowledge”?
Response: We have rephrased the sentence.
R3. 2. Line 62. At the same time?
Response: It was corrected.
R3. 3. Line 64-35. The meaning of “biological point of view” is unclear. In the above parts of the MS, the features of the Antarctic plants explained at the biological point of view. Please be more specific. Line 24-25 should be also changed.
Response: Sorry, but in the text, we specify the biological aspect not yet fully studied in these plants. “For instance, factors that promote/restrict their distribution at different spatial scales, the genetic base associated with the stress resistance, biochemical traits as saturated and unsaturated fatty acid ratio and the production and allocation of non-structural carbohydrates, are among the biological aspects not yet fully studied in these plant species” and we refer to these aspects.
R3. 4. Line 89. The meaning of “different” is unclear.
Response: It was changed for a “variety”
R3. 5. Line 109. Please add an explanation for LT50.
Response: We have included in the text an explanation for LT50.
R3. 6. Line 112. What is the meaning of “complete individuals without nucleating agents”?
Response: It was changed as whole plant. It is part of the methodology used by Gianoli et al. (2004) to determine freezing resistance (LT50). They evaluated freezing resistance by the survival of plants after freezing at a given temperature without using nucleating agents, for example.
R3. 7. Line 117, Table 1. Please add explanation for the condition of cold acclimation.
Response: We have included in Table 1 an explanation for the condition of cold acclimation.
R3. 8. Line 130. Please insert line break before “Different studies”.
Response: It was corrected.
R3. 9. Line 133. What is the meaning of “wide cellular distribution”?
Response: We have eliminated this sentence.
R3. 10. Line 137. What is the meaning of “negative carbon balance”?
Response: It means when the respiratory rate exceeds the photosynthetic rate due to a metabolic shift induced by environmental stressors. A sentence clarifying this issue was included.
R3. 11. Line 150-151 and Line 154-160. These parts seem to be inconsistent. The former sentence stated that accumulation of sugar could be caused by reduction in growth. However, the latter sentences explained that high sucrose concentration is related to high activity of sucrose phosphate synthase, which can provide carbohydrate to supply rapid growth.
Response: It was corrected.
R3. 12. Line 168. What is the meaning of “among them”?
Response: It was rephrased in the text.
R3. 13. Line 184. Under high light?
Response: It was corrected.
R3. 14. Line 191. What is the meaning of “despite species specific”?
Response: It was re phrased.
R3. 15. Line 207-209. I cannot understand this sentence.
Response: It was re phrased.
R3. 16. Line 213-245. I cannot understand this part. How are these explanations related to adaptation to Antarctic environment?
Response: A sentence has been included in this part of the manuscript.
R3. 17. Line 238 – 242. This part is complicated. How is the relationship between ETR and CaOx crystals?
Response: We rephased the sentence to clarify the relationship between ETR and calcium oxalate crystals. In brief, the hypothesis of the alarm photosynthesis mechanism suggests that, under specific stress conditions, plants can utilize endogenous CO2 released during the degradation of calcium oxalate crystals. ETR has served as an indirect measurement to demonstrate that, in stress or CO2-limiting conditions, plants maintain an electron flow that may be linked to photosynthetic activity.
R3. 18. Line 250 – 255. I cannot understand the difference between Antarctic species and non-Antarctic species.
Response: A sentence was included to clarify the use of “non-Antarctic species”.
R3. 19. Line 262. What are “Unknown-03 and Unknown-06”?
Response: They are metabolites that have been detected but its specific chemical nature or structure has not yet been determined or is not cataloged in the database used for the analysis. In the text it is mentioned that they are metabolites.
R3. 20. Line 286-325. How are these explanations related to adaptation to Antarctic environment? The authors stated that Antarctic plants have high capacities to cope with photoinhibition (line 329-330), so oxidative stress may hardly occur.
Response: Yes, both Antarctic plant species have photoprotective mechanisms. Antioxidant system is part of these mechanisms. Actually, D. antarctica can use oxygen as an electron alternative sink without suffering any damage because of its active antioxidant enzymatic machinery. So, a high photoprotective capacity does not exclude an efficient antioxidant system, is part of it.
R3. 21. Line 331. Different from what?
Response: We have rephased the sentence.
R3. 22. Line 376. Please use lowercase 3 for C3.
Response: It was corrected.
R3. 23. Line 376-401. It is better to move this part to the section of “2.2 photosynthesis (line 181)” and rearrange the sentences.
Response: We don’t agree with this suggestion because the whole paragraph is about photorespiration and photoprotective mechanisms. So, we believe it is well placed in section 2.4 Photoprotective mechanisms.
R3. 24. Line 427. Thinner cross section => thicker cross section?
Response: It was a mistake; it is now corrected in the manuscript.
R3. 25. Line 428. What is the meaning of “narrow abaxial surface”?
Response: It refers to the fact that the leaves have a shorter adaxial surface length, i.e., smaller leaves.
R3. 26. Line 431. What is the meaning of "habit size”?
Response: It refers to the overall growth form or structure of the plants. It includes characteristics such plants height, width, and overall visual appearance. A sentence clarifying this issue was included.
R3. 27. Line 451-452. Light response curves?
Response: It was corrected.
R3. 28. Line 486-488. The authors explained physiological mechanisms in the above section in detail. Please be more specific.
Response: The mechanisms were mentioned.
R3. 29. Line 494-612. This part is highly descriptive and hard to read. Please rearrange the sentences, and use the figure 1 in this part.
Response: Thank you for this suggestion, the Fig. 1 mention was incorporated in the section 4, especially in the Photosynthesis and Respiration responses to warming.
R3. 30. Line 614-615. Where is the explanation for the xeromorphic anatomical characteristics?
Response: It was corrected. We have added LMA and LD as xeromorphic parameters.
R3. 31. Table 2. There is no need using a table. Please describe in the text.
Response: Table 2 was deleted, and questions were incorporated in the text.